# Human CD4^+^CD45RA^+^ T Cells Behavior after In Vitro Activation: Modulatory Role of Vasoactive Intestinal Peptide

**DOI:** 10.3390/ijms23042346

**Published:** 2022-02-20

**Authors:** Raúl Villanueva-Romero, Alicia Cabrera-Martín, Emigdio Álvarez-Corrales, Mar Carrión, Selene Pérez-García, Amalia Lamana, David Castro-Vázquez, Carmen Martínez, Rosa P. Gomariz, Irene Gutiérrez-Cañas, Yasmina Juarranz

**Affiliations:** Departamento de Biología Celular, Facultad de Biología y Facultad de Medicina, Universidad Complutense de Madrid, 28040 Madrid, Spain; ravillan@ucm.es (R.V.-R.); alicab02@ucm.es (A.C.-M.); emigdalv@ucm.es (E.Á.-C.); macarrio@ucm.es (M.C.); selene@ucm.es (S.P.-G.); amaliala@ucm.es (A.L.); dcastr01@ucm.es (D.C.-V.); cmmora@ucm.es (C.M.); gomariz@ucm.es (R.P.G.); irgutier@ucm.es (I.G.-C.)

**Keywords:** naїve Th cells, CD4^+^CD45RA^+^ cells, Vasoactive intestinal peptide, VPAC receptors

## Abstract

Naїve CD4^+^ T cells, which suffer different polarizing signals during T cell receptor activation, are responsible for an adequate immune response. In this study, we aimed to evaluate the behavior of human CD4^+^CD45RA^+^ T cells after in vitro activation by anti-CD3/CD28 bead stimulation for 14 days. We also wanted to check the role of the VIP system during this process. The metabolic biomarker Glut1 was increased, pointing to an increase in glucose requirement whereas Hif-1α expression was higher in resting than in activated cells. Expression of Th1 markers increased at the beginning of activation, whereas Th17-associated biomarkers augmented after that, showing a pathogenic Th17 profile with a possible plasticity to Th17/1. Foxp3 mRNA expression augmented from day 4, but no parallel increases were observed in IL-10, IL-2, or TGFβ mRNA expression, meaning that these potential differentiated Treg could not be functional. Both VIP receptors were located on the plasma membrane, and expression of VPAC_2_ receptor increased significantly with respect to the VPAC_1_ receptor from day 4 of CD4^+^CD45RA^+^ T activation, pointing to a shift in VPAC receptors. VIP decreased IFNγ and IL-23R expression during the activation, suggesting a feasible modulation of Th17/1 plasticity and Th17 stabilization through both VPAC receptors. These novel results show that, without polarizing conditions, CD4^+^CD45RA^+^ T cells differentiate mainly to a pathogenic Th17 subset and an unpaired Treg subset after several days of activation. Moreover, they confirm the important immunomodulatory role of VIP, also on naїve Th cells, stressing the importance of this neuropeptide on lymphocyte responses in different pathological or non-pathological situations.

## 1. Introduction

CD4^+^ T (Th) cells play a pivotal role in immune response orchestration and are responsible for an effective and appropriate adaptive immune response to pathogens. Peripheral CD4^+^ T cells populations are heterogeneous, comprising naїve, effector, memory, and terminally differentiated effector cells (TEMRA). Forty-five percent of CD4^+^ T cells from peripheral blood are positive for CD45RA marker, forty percent are naїve cells, and the other five percent are TEMRA cells [1]. Therefore, CD45RA is a conventional marker for naїve Th cells, although there are other markers to identify and classify them [2]. Once they come out of the thymus, naїve Th cells constantly recirculate between secondary lymphoid organs, the lymph and the blood, where they differentiate to effector and memory Th cells upon physical interaction with antigen-presenting cells (APC), optimizing the efficiency of the adaptive immune response [3]. Naїve CD4^+^ cells are also a heterogeneous population, and it can be distinguished into two important subsets, the youngest naїve T cells (CD31^+^) and mature naїve T cells (CD31^−^) [2,4]. The proportion of the CD31^+^ subset in peripheral blood decreases with aging, in parallel with thymic involution, whereas the CD31^−^ subset remains relatively constant throughout adult life, leading to a progressive rise in the proportion of CD31^−^ within the naїve CD4^+^ compartment [4].

Cellular development, activation, differentiation, function, and survival are intrinsically linked to metabolic remodelling. After antigen presentation and their subsequent activation, naїve T cells increase their demand for ATP, leading to a boost in glycolytic and mitochondrial metabolism. Thus, the rise in glucose transporter type 1 (Glut1) expression and regulation of its trafficking are very important in the function of Th cells [5].

During T cell receptor (TCR) activation, naїve T cells are exposed to different polarizing signals that induce expression of different lineage-specific transcription factors and result in the generation of distinct Th subsets such as effector/memory Th cells, which comprise not only Th1, Th17, Th2, Th22, Th9 and follicular Th cells (Tfh) but also regulatory Th cells (Treg). Each Th subset has characteristic functional properties, such as cytokine production and chemokine receptor expression, to induce the correct adaptive immune response [6,7]. During the differentiation pathway towards effector/memory Th or Treg cells, the mediators present in the microenvironment of CD4^+^CD45RA^+^ cells are essential to acquire lineage commitments. However, these lineage commitments are not terminal or locked, and they could be changed, allowing T cells to switch their phenotype towards mixed or alternative fates in response to changing environments and situations—what we know as T cell plasticity. Among all Th subsets, Th17 are of the most heterogeneous and plastic population, having two different subsets described, named non-pathogenic Th17 and pathogenic Th17 [8,9,10,11]. Besides, several reports show their capability to transform into other subsets, such as the non-classical Th1 ex-Th17 cells [6,9,11,12]. The heterogeneity of Th cells helps target the cells to the tissues where they are required and outline the class of immune and tissue response that is proper for the type of pathogen attack. However, some of these subsets have been related to a greater or lesser degree with malfunctions of the immune system, specifically with autoimmune diseases such as Th1 and Th17, which are closely related to rheumatoid arthritis, Crohn’s disease, etc. [6,7,11].

Beyond its influence on plasticity and the phenotype change of effector/memory cells, the molecular composition of the microenvironment might even affect the phenotypic characteristics of naїve populations prior to antigen presentation, enhancing tonic signalling but also favoring inhibitory pathways related to cytokine-induced signalling [13]. Changes in T cell-intrinsic factors related to aging have been reported in both naїve CD8 and CD4 T cell compartments. For instance, it has been described as having better chromatin accessibility for transcription factors related to BAFT family (AP-1, c-Jun) in naїve CD8 T cells [14]. Moreover, in naїve CD4 T cells, attenuated TCR signalling in older subjects promotes Th2 development [15].

As mentioned above, the microenvironment of naїve T cells is critical in controlling their activation and differentiation process. Neuropeptides such as vasoactive intestinal peptide (VIP) are found in this microenvironment as they are secreted by different cells, including lymphocytes themselves, and they even arrive through nerve endings [16,17]. VIP is known as an anti-inflammatory and immunomodulatory neuropeptide that balances and modulates the pathogenic activity of various cellular subpopulations of the immune system, including CD4^+^ T lymphocytes. In vivo and in vitro studies with different mice autoimmune models, human samples from healthy donors and rheumatoid arthritis patients have demonstrated that VIP is able to decrease pathogenic Th1 and Th17 subsets, and favor Th2, Treg and non-pathogenic Th17 subsets. Indeed, VIP regulates the plasticity between these subsets [17,18,19], but no study has demonstrated the VIP’s role in any preconditioned microenvironment activation of human CD4^+^CD45RA^+^ T cells. This neuropeptide exerts its actions through two specific receptors named VPAC_1_ and VPAC_2_ that belong to the B1 family of G-protein coupled receptors (GPCRs). In different non-immune and immune cells, including fibroblast synoviocytes, in vitro differentiated osteoclast, macrophages and lymphocytes, the pattern expression of these receptors changes with the degree of cellular activation, maturation or differentiation [17,20,21,22]. In the specific case of human and mouse CD4^+^ T lymphocytes, it was described that their activation induces a loss in mRNA expression of VPAC_1_, whereas an important increase in VPAC_2_ appears [17,23,24,25]. Moreover, in human-activated or senescent memory Th cells from healthy donors or early arthritis patients, protein expression of VPAC_2_ was increased, whereas VPAC_1_ did not change. Changes have been observed not only in the expression pattern of these receptors with the activation or maturation of Th lymphocytes, but also in their cellular location (Villanueva-Romero et al., 2019; Villanueva-Romero et al., 2020). For example, in resting memory Th cells, VPAC_1_ was located on the plasma membrane and nucleus, whereas it only appeared in the nucleus after cellular activation. On the other hand, the VPAC_2_ receptor was always found in the plasma membrane location. From a translational point of view, the VIP axis has an important potential in the control of inflammation and the modulation of several autoimmune diseases, but it could also be used as a biomarker for personalized treatment in autoimmune diseases such as rheumatoid arthritis [17,26,27,28].

Given the lack of studies on the behavior of CD4^+^CD45RA^+^ T cells during in vitro activation for short and long periods of time without any Th-lineage polarization microenvironment, we decided to analyze in these in vitro conditions the behavior of these cells from a metabolic and differentiation point of view. Furthermore, given the important functions of the VIP axis in the immune system and its involvement in inflammatory and autoimmune diseases, we also aimed to study this axis during the activation of naїve T cells under these in vitro conditions.

## 2. Results

### 2.1. CD4^+^CD45RA^+^ Cells Viability during Activation

CD4^+^CD45RA^+^ T cells were activated in presence of CD3/CD8 antibodies and their viability were monitored. There was an increase in the number of cells observed from day 4 and 7, but it decreased after 10 days of culture in line with the decreasing viability followed (Table 1).

### 2.2. Metabolic Biomarkers in CD4^+^CD45RA^+^ Cells during Their Activation

Metabolism is intrinsically linked to CD4^+^CD45RA^+^ T cells’ activation, differentiation, function, and survival. After activation, these cells increase their demand of ATP, so an increase in glycolytic and mitochondrial metabolism is performed. We tested two important molecules involved in metabolism, Glut1 and hypoxia-inducible factor 1 alpha (Hif-1α) (Figure 1). The glucose transporter Glut1 is necessary for the activation and differentiation of these cells, whereas Hif-1α is a key metabolic sensor. GLUT1 mRNA expression augmented significantly until day 4 of activation and decreased thereafter; however, HIF-1α was higher in resting cells than activated cells. This means that in the first days of activation, high glucose requirement results in an increased Glut1 gene expression, and normoxia culturing conditions constrain a decreased HIF-1α expression.

### 2.3. Functional Pathogenic Th17 Differentiation Profile Is the Main Subset That Appeared after Four Days of CD4^+^CD45RA^+^ Cells Activation

Once activated, CD4^+^CD45RA^+^ T cells can differentiate into different Th subtypes, depending on the stimulus, microenvironment, APC, and pathological situation. Our experiments tried to essay the natural differentiation Th profile of CD4^+^CD45RA^+^ T cells, mimicking the activation via TCR without any preconditioned microenvironment to force the different Th subsets. First, we analyzed the differentiation into pathogenic effector Th cells. No differentiation profile towards Th2 subsets was observed with CD4^+^CD45RA^+^ in these activation conditions (data not shown). The expression of the Th1 subset-specific transcription factor, Tbx21 (T-bet), increased on the first day of activation and then decreased significantly on the remaining days (Figure 2A). In contrast, the expression of the Th17 subset-specific transcription factor, RORC (RORƴ), significantly increased progressively from day 1 and remained stable after 7 days of culture. When comparing different Th1- (Tbx21, STAT1, IFNγ, IL-2, TNFα) with Th17-associated biomarkers (RORA, RORC, STAT3, IL-10, IL-9, IL-22, IL-23R), we observed an increase in those associated to Th1 on day 1, whereas Th17-associated biomarkers were increased on day 7 (Figure 2B). The mRNA expression of the IL-23 receptor, the cytokine necessary for the stabilization of Th17 subset, increased significantly from day 1 of activation, remaining stable during the rest of the culture days (Table 2). These results indicate that without any preconditioned medium, the differentiation of CD4^+^CD45RA^+^ T cells to Th1 only appreciated at short times of cell activation while the differentiation to Th17 appeared progressively from day 4 of activation.

In view of these results, we decided to analyze the nature of these Th17 cells. IL-22 is a cytokine characteristic of a pathogenic Th17 phenotype, whereas IL-9 and IL-10 are characteristic of a non-pathogenic Th17 phenotype. The Scatter Plots in Figure 2B indicate that IL-10 and IL-9 appeared increased on day 1 with respect to the resting situation whereas this rise was reversed on day 7, when both IL-10 and IL-9 decreased their expression compared to day 1. To go in depth into this fact, we performed IL-22/IL-9 and IL-22/IL-10 ratios at mRNA level, and upon observing that both increased on day 4 and 7 but diminished after that (Figure 3A), we corroborated the same behavior for IL-22/IL-10 ratio at protein level (data not shown). Pathogenic Th17 can also produce IFNγ and TNFα, contributing to reprogram cells to Th1 profile, and in this sense, IFNγ mRNA expression augmented significantly from day 7, reaching a 6.5-fold increase on day 14 versus day 7. (Figure 3B) In protein level, IFNγ also showed the highest expression on day 1 of culture (data not shown). The mRNA expression of TNFα and Tbx21 was also augmented slightly in this period, reaching 2.3- and 1.8-fold higher levels on day 14 than day 7, respectively (Figure 2A and Figure 3B). This data indicate that the proportion of pathogenic Th17 was higher than non-pathogenic Th17 during the differentiation of CD4^+^CD45RA^+^ T cells after several days of activation. Indeed, pathogenic Th17 differentiated cells could acquire a Th17/1 profile from day 7.

After checking the effector Th profile of these cells, we tested the expression of the master regulator of the Treg subset, Foxp3, and its characteristic cytokine, IL-10 (Figure 4). Foxp3 expression augmented significantly on day 4 and remained at these levels for the rest of the activation period. However, the expression of IL-10 and TGFβ cytokines associated with functional Treg, and IL-2, the cytokine necessary for Treg stabilization, increased on day 1 and decreased drastically thereafter. We also observed this behavior in the protein level, observing higher expression on day 1 of culture, which decreased thereafter (Figure 4). This means that although CD4^+^CD45RA^+^ T cells can spontaneously in vitro differentiate to Treg in these activation conditions, these cells are not functional because IL-10, TGFβ, and IL-2 levels do not increase their expression together with Foxp3.

### 2.4. The Expression Pattern and Cellular Localization of VPAC Receptors Change in Human Activated CD4^+^CD45RA^+^ Cells

VPAC_1_ and VPAC_2_ are the main VIP receptors that mediate many immune functions in T cells. Figure 5A shows that VPAC_1_ mRNA expression decreased during the first seven days of activation and augmented afterward, whereas VPAC_2_ mRNA expression increased significantly from day 4 of activation onwards. Thus, the presence of the VPAC_2_ receptor increased significantly with respect to VPAC_1_ receptor on day 4 of CD4^+^CD45RA^+^ T activation. Through western blot analysis, we observed that there were no substantial changes in VPAC_1_ expression at protein level, whereas VPAC_2_ expression increased significantly after seven days of activation (Figure 5B).

The fluorescence intensity and orthogonal view of the immunofluorescence staining studies corroborated the presence of both VPAC receptors in resting and activated CD4^+^CD45RA^+^ T cells (Figure 6). Both VPAC receptors are expressed in plasma membrane location without showing any difference of distribution between resting and activated CD4^+^CD45RA^+^ T cells. To further study their expression, we carried out a distribution analysis by fluorescence intensity and 3D visions, which confirmed our previous observations.

### 2.5. VIP Decreases Th17/1 Plasticity and Pathogenic Th17 Stabilization in Activated CD4^+^CD45RA^+^ Cells

After studying the presence of VPAC receptors and their behavior during the activation of conventional human CD4^+^CD45RA^+^ T cells, we analyzed the response of these cells to VIP presence, considering that our stimulation mimics the activation via TCR without any preconditioned microenvironment. VIP changed neither the viability of these cells nor the metabolic profile (data not shown). Regarding the differentiation profile, VIP decreased IFNγ and IL-23R mRNA expression during the differentiation of these cells without affecting other differentiation markers (Table 2 and data not shown). This effect was also observed when the cells were activated in the presence of both specific VPAC agonists, pointing to a role of VIP reducing the possibility of Th17/1 plasticity and stabilization of the pathogenic Th17 generated after CD4^+^CD45RA^+^ T cells activation and, probably through both VPAC receptors signalling.

## 3. Discussion

The importance of understanding the behavior of naїve CD4^+^ T cells, their response during an activation process, and their differentiation profile is vital to understanding the immune response. More than that, it is essential to comprehend that different mediators present in the immune microenvironment might leave a mark on, and modulate naїve populations, and how this could be beneficial for its possible translation to immunological disorders such as autoimmune diseases. In this sense, the neuropeptide VIP is an important anti-inflammatory and immunomodulatory mediator that could affect the behavior of human naїve CD4^+^ T cells. So far, to our knowledge, the role of VIP has been studied in total human CD4^+^ T cells, or in the memory CD4^+^ T cells, but there are no important studies describing its role or the expression of its receptors⁠—VPAC_1_ and VPAC_2_⁠—in naїve cells during their activation process and without any differentiation polarization. Therefore, with this study, we have tried to fill this gap in the knowledge of the VIP axis in the immune system with the intention of trying to apply this knowledge from a translational point of view in future research.

Even though there is no unique naїve CD4^+^ T marker in humans, we can assume that almost 90 percent of CD45RA^+^CD4^+^ T cells in periphery blood are naїve T cells [1,2], and thus we used this marker to select human naїve CD4^+^ T cells. In addition, there is increasing evidence showing that naїve T cells are heterogeneous in phenotype, function, dynamics, and differentiation status. It has been described in the existence of two main subtypes of human naїve T cells: the youngest, or CD31^+^, and mature, or CD31^−^, naїve T cells. With age, the number of naїve CD4^+^ T cells decreases, but also their heterogeneity, as the CD31^+^ subset decreases considerably [2,4]. In consequence, we assume that the majority subset present in our blood samples from middle-aged donors is the mature, or CD31^−^, naїve T subset.

Before considering the role of VIP and VPAC receptors during the activation/differentiation of human CD45RA^+^CD4^+^ T cells, we studied the behavior of these cells activated in non-polarizing conditions. Since proper metabolism is a prerequisite for naїve cells to become active and function properly, we first studied the expression of two key metabolic mediators in this process, Glut1 and Hif-1α [5]. Peripheral naїve T cells use mainly fatty acid oxidation to fuel tricarboxylic acid (TCA) and oxidative phosphorylation to generate ATP and import small amounts of glucose [5,29]. The activation of these cells through TCR-mediated signals induces an increase in both glycolytic and mitochondrial metabolisms that are essential for T cell activation [5]. Under our cell activation conditions, *GLUT1* mRNA expression increased significantly until day 4 of activation and decreased thereafter but remained at higher levels than in resting cells. This fact indicates that CD4^+^CD45RA^+^ cells need large amounts of glucose in the early stage of activation/differentiation, while in longer stages of activation, glucose requirements decrease but remain higher than in a resting state. The mRNA expression of the other metabolic marker, *HIF-1α*, was higher in resting cells than in activated cells. This is probably related to the fact that our in vitro activation condition did not mimic physiological O_2_ condition, as O_2_ tension is generally low in secondary lymphoid tissues compared to the bloodstream or atmosphere and, therefore, the presence of Hif-1α is more necessary to increase the glycolytic metabolism [30,31]. According to our results, it has been described that overexpression of Glut1 in T cells preserves effector functions, but they do not terminally differentiate as they persist and become long-term effector memory T cells [31]. On the other hand, overexpression of Hif-1α on CD4^+^ cells enhance Th17 proportion [6]. Moreover, the metabolic phenotype varies between the distinct CD4^+^ T cell subsets [5], and therefore, our results are supported by the differences observed between T cell populations.

One of our main objectives was to determine which CD4^+^CD45RA^+^ T cell subsets differentiate spontaneously when there is no Th polarizing condition in the culture medium, except for mimicking TCR activation. We found no Th2 differentiation, whereas Th1 profile was observed in early stage of activation and pathogenic Th17 subset appeared after four days of activation. We came across a spontaneous Treg differentiation, but these regulatory Th cells were not functional. As we have mentioned above, our CD4^+^CD45RA^+^ T cells are mostly mature naїve T cells, and as previous results showed that naїve CD4^+^ T cells from young individuals are more likely to differentiate towards Th2 subset, it is not surprising that we found no Th2 differentiation [32]. On day 1 of cell activation/differentiation, several markers linked to a Th1 phenotype such as IFNγ, T-bet, STAT1, TNFα, IL-2 and IL-10 increased in relation to Th17-markers. The first three molecules are characteristic of Th1 subsets, but TNFα is also a proinflammatory cytokine that acts as a co-stimulatory molecule for T-cells, increasing the activation and proliferation of T cells through NFκB signalling pathways [33]. That explains why our results show a TNFα peak on day 1 but decrease in the following days. Although a slight increase was also observed after day 10 of culture, it could have been because pathogenic Th17 cells can shift their commitment towards Th1 subsets producing Th1-associated markers. Indeed, it has been described the existence of a Th17/1 mixture subset exhibiting a IFNγ or T-bet profile and that coexists with high levels of the master regulator of Th17 subset, RORγ [6,9,11,12]. IL-2 is a pleiotropic cytokine that controls the differentiation and homeostasis of both pro- and anti-inflammatory T cells. While this cytokine antagonizes Th17 differentiation, Th1 and Treg need it for their cellular function and maturity [34]. These statements fit well with the IL-2 profile observed in CD4^+^CD45RA^+^ T cells after activation, and with our hypothesis that Th1 and Treg cells develop at early stages, and once IL-2 expression decreases, they give way to Th17 differentiation and non-functional Treg cells. Like IL-2, IL-10 is a cytokine originally associated with Th2 and Treg subsets [35], but it can also be produced by Th1 subsets to limit the collateral damage caused by exaggerated inflammation or Th17 cells when they show a non-pathogenic phenotype [6,9,11,12,36,37]. Our results showed a significant peak in IL-10 expression on day 1 of CD4^+^CD45RA^+^ T cells activation and remained practically at zero for the rest of the culture days. This could be explained by three events: the cells that differentiate to Treg are non-functional; there is no differentiation to Th1 from naїve T cells since day 1; or Th17 that are generated since day 4 are pathogenic. It is important to note that no spontaneous Th2 differentiation was observed in our culture conditions. All the above-mentioned events as well as the balance between cytokines associated with a pathogenic Th17 profile (IL-22) or non-pathogenic Th17 profile (IL-9 and IL-10) indicate that in vitro activated CD4^+^CD45RA^+^ T cells take a lineage commitment towards Th17 pathogenic profile. This Th17 subtype shows plasticity towards a Th17/1 profile and is closely associated with autoimmune diseases [9,38,39]. It has been described in several autoimmune diseases that Th17 is liable to initiate the disease, but Th17/1 plasticity is responsible for perpetuating it [38,40,41].

Regarding Foxp3 levels, we can assume that CD4^+^CD45RA^+^ T cells differentiate spontaneously to Treg during their activation in vitro from day 4. This process could reproduce the natural peripheral Treg (pTreg) differentiation from conventional CD4^+^ T cells. These cells need the influence of TGFβ and IL-2 to acquire their lineage commitment and are less stable than thymic Treg. Indeed, they produce IL-10 and TGFβ to maintain self-tolerance and suppress the activity of effector Th cells via IL-10 and TGF-β production, and through cell–cell interactions [42,43,44]. Present results indicate that although Foxp3 expression increased from day 4, there was practically no IL-10 or IL-2 expression from day 4 to 14. In addition, TGFβ expression decreased significantly in these cultures. These results suggest that even if pTreg cells spontaneously differentiate under these activation conditions, they are not functional. This fact can also be observed in several pathological conditions where several factors control the functionality of Treg cells [44,45].

As we mentioned before, the microenvironment can modulate or generate some susceptibilities on naїve populations prior to TCR activation. Just a few studies have reported features or changes in naїve T cell-inherent factors or lineage bias, most of them from an age-related point of view. For instance, work in mice models has reported Th2 development via DUSP6 and SIRT1 upregulation in elderly individuals, which leads to impairment in TCR activation [15]. In human studies, naїve T cells from elderly people show enhanced development of Th17 cells in response to TCR triggering, together with Th17-polarizing mediators compared to cells from young people [46], whilst recent studies show that naїve cells from individuals older than 60 years are susceptible to acquiring a Th9 profile [47]. Even though there were procedural differences between our research and those experiments (where naїve T cells were activated under polarizing conditions), we can conclude that systemic conditions such as aging, frequently named inflame-aging, play a role in cell pathways before classical activation. This imprint could be relevant in the study of autoimmune diseases, where a network of diverse cytokines is produced, and constitutes a “cytokine milieu” that drives chronic inflammation. In rheumatoid arthritis patients, chronic exposure of total T cells to TNFα impairs cell-mediated immune responses via CD28 downregulation [48] whereas memory CD4^+^CD45RO^+^ T cells acquire a stronger Th17 pathogenic profile compared to cells from healthy donors under Th17-polarizing conditions [49].

After analyzing the behavior of human CD4^+^CD45RA^+^ T cells in our culture conditions, we studied VIP axis during the activation of these cells. In the puzzle of the data describing the role of VIP and its receptors in the immune system [17,18,19,50], a key piece is missing: understanding how the presence of VIP affects the activation/differentiation of human naїve T cells without any preconditioning culture medium, and how its receptors are modulated during this process. To fill this gap, we first analyzed the expression of VPAC receptors, observing a significant increase in VPAC_2_ mRNA and protein expression after 4 days of activation similar to that observed in other lymphocyte states as the activated CD4^+^CD45RO^+^, senescent lymphocytes or Th17-differentiated cells [17,18,22,51,52]. There is a difference in the timing of memory Th cells activation, as the increase in VPAC_2_ receptor expression was observed from day 1 in this last population [52]. This difference could be due to the intrinsic characteristic of each Th subpopulation. Some studies evince that the balance between VPAC_1_ and VPAC_2_ receptors, in pathological conditions or when immune system cells are activated, is always in favor of VPAC_2_, suggesting that this receptor could play a key role in these situations [17,19]. This statement further correlates with the fact that the expression of both receptors has been proposed as a biomarker in some autoimmune diseases, such as early arthritis or Graves’ disease [53]. VIP actions can be affected not only by changes in VPAC expression in lymphocytes but also changes in cellular location as occurs with other GPCRs or even VPAC_1_ in other types of cells [54,55,56,57]. In resting human memory (CD4^+^CD45RO^+^) or senescent Th lymphocytes (CD4^+^CD28^−^), VPAC_1_ is located on the plasma membrane and nucleus whereas it only appears in the nucleus in activated memory Th cells [22,52]. Present results with CD4^+^CD45RA^+^ T cells show similar cellular location in resting and activation states, where both VPAC_1_ and VPAC_2_ receptors are always found in the plasma membrane location independently of activation state as occurs with memory Th cells [52]. Goetzl hypothesized that VPAC receptors constitute a dynamic system for signalling in T cells, predicting that responses in the plasma membrane location would have a fast onset and brief duration, whereas receptors in the nuclear membrane would have responses with slow onset and sustained in time, as has been observed for other GPCRs [58,59,60]. Moreover, other GPCRs receptors, such as metabotropic glutamate receptor 5 and TSH receptor, trigger different cellular responses depending on their cellular location [61,62]. Therefore, it cannot be ruled out that changes in the cellular location of VPAC_1,_ depending on the activation state of the lymphocytes, could generate different functional responses in the cells. To test this possibility and study VIP modulation of activation/differentiation profile of CD4^+^CD45RA^+^ T cells, we analyzed all markers related to metabolism and differentiation of T cells in the presence of VIP and specific VPAC agonists. The presence of VIP did not change any marker tested apart from IL-23R and IFNγ, which decreased in the presence of VIP and both specific VPAC agonists. IL-23R is a receptor essential for Th17 stabilization and expansion [6,35] and IFNγ is a Th1 cytokine that can also be produced by pathogenic Th17 and Th17/1 subsets [6,9]. Thus, the presence of VIP during the activation/differentiation of CD4^+^CD45RA^+^ T cells without any preconditioning medium could decrease Th17 cell stabilization/expansion and pathogenicity by lessening Th17/1 plasticity, through both VPAC receptors. Other studies carried out during the activation of human memory Th cells from healthy donors and early arthritis patients, without any preconditioning medium, corroborate these findings [39,52]. Furthermore, the VIP and both the VPAC receptors have an immunomodulatory effect. When memory Th cells from healthy donors and early arthritis patients are differentiated in Th17 polarizing conditions in the presence of TGFβ, which induces a non-pathogenic profile, the presence of VIP further decreases the pathogenic Th17/1 profile and increases Th17/Treg profile [49]. In addition, during the differentiation of human naїve T cells from healthy donors to Th17 cells in these last culture conditions, opposite effects on IL-23R were observed mainly through the VPAC_2_ receptor [51]. Taking everything into consideration, we could take a stand that the effects of VIP on activation/differentiation of human memory CD4^+^CD45RO^+^ T cells are greater than those on naїve CD4^+^CD45RA^+^ T cells. Moreover, when the latter are induced to differentiate towards a non-pathogenic Th17 profile, the presence of VIP and its agonists favors this differentiation.

## 4. Materials and Methods

### 4.1. Healthy Donors

Samples from 12 healthy donors were included in this study. The study was performed according to the recommendations of the Declaration of Helsinki and was approved by the Ethics Committees of the Transfusion Center of CAM. Healthy donor samples were obtained from buffy coats from the Transfusion Center. Following the Spanish Personal Data Protection law, the patients’ demographic information was kept confidential. All patients signed an informed consent form before sampling.

### 4.2. Isolation of Human Peripheral Blood CD4^+^CD45RA^+^ T Cells

CD4^+^CD45RA^+^ T cells were isolated from whole blood from healthy donors. For mononuclear cell isolation, density gradient centrifugation by Ficoll–Hypaque (Merck KGaA, Darmstadt, Germany) was performed. CD4^+^ T cells were isolated by RosetteSep™ Human CD4^+^ T Cell Enrichment Cocktail (Stem Cell Technologies, Vancouver, Canada). CD4^+^CD45RA^+^ T cells were then isolated by negative selection using an EasySep™ Human Naїve CD4^+^ T Cell Isolation Kit (Stem Cell Technologies, Vancouver, Canada). The purity of CD4^+^CD45RA^+^ T cells was greater than 92%.

### 4.3. In Vivo Activation of Human CD4^+^CD45RA^+^ T Cells

CD4^+^CD45RA^+^ T cells were cultured in 96 U-well plates (0.1 × 106 cells/well), with RPMI-1640-GlutaMAX media (Life Technologies, Carlsbad, CA, USA) supplemented with 10% FBS (Lonza, Basel, Switzerland) and 1% penicillin/streptomycin (Life Technologies, Carlsbad, CA, USA). Cells were activated/expanded with Dynabeads Human T-Activator CD3/CD8 (Life Technologies, Carlsbad, CA, USA). CD4^+^CD45RA^+^ T cells were cultured in the absence (activation condition) or presence of 10nM of VIP (Bachem A.G., Bubendorf, Switzerland), VPAC_1_ agonist (K15R16L27VIP (1-7)/GRF (8-27)) or VPAC_2_ agonist (RO 25-1553) (Bachem A.G., Bubendorf, Switzerland) during different time points (1, 4, 7, 10 and 14 days). Cells without activation or any treatment and collected on day 0 were considered resting cells. Cell viability and concentration was determined by an Automated Cell Counter, EVE™ (NanoEnTek Inc, Seoul, Korea).

### 4.4. RNA Extraction and Semiquantitative Real-Time PCR

For total RNA extraction, we used the TriReagent method (Merck KGaA, Darmstadt, Germany). RNA (2 µg) was reverse transcribed using a High Capacity cDNA Reverse Transcription Kit (Life Technologies, Carlsbad, CA, USA). Semiquantitative RT-PCR analysis for all molecules tested (Table 3) was performed using TaqMan Gene Expression Master Mix (Applied Biosystems, Waltham, MA, USA). We normalized each sample with succinate dehydrogenase complex flavoprotein subunit A (SDHA), using the formula 2^−ΔCt^. Amplification was performed in a 7900HT Fast Real-Time PCR System apparatus (Applied Biosystems, Waltham, MA, USA).

### 4.5. ELISA

Protein levels of IL-22, IL-10, IFNγ and TGF-β were measured in culture supernatants by commercial ELISA kits (Invitrogen, Waltham, MA, USA), according to the manufacturer’s instructions.

### 4.6. Western Blot

Protein extracts were obtained in an ice-cold RIPA buffer. Protein extracts (10 and 40 µg for VPAC_1_ and VPAC_2_, respectively) were subjected to sodium dodecyl sulfate polyacrylamide gel electrophoresis (SDS-PAGE) and blotted on a polyvinylidene difluoride (PVDF) membrane (Bio-Rad Laboratories, France). After blocking, membranes were incubated overnight at 4 °C with rabbit polyclonal anti-human VPAC_1_ antibody (1:10,000, Thermo Fisher Scientific) and mouse monoclonal anti-human VPAC_2_ antibody (1:1000, Abnova, Tapei, Taiwan). Mouse monoclonal anti-beta actin (ACTB) (1:10,000, Merck KGaA, Darmstadt, Germany) was used as loading control. Appropriate horseradish peroxidase-conjugated secondary antibodies (1:10,000, Santa Cruz Biotechnology, Dallas, TX, USA) were applied and detected by Pierce SuperSignal West Pico (Thermo Scientific, Waltham, MA, USA). Protein bands were analyzed using the Bio-Rad Quantity One program and normalized against β-actin.

### 4.7. Immunocytochemistry Staining

Cell suspensions (resting and activated cells on day 7) were seeded in PBS 1x and held onto SuperFrost Plus slides (Thermo Scientific, Waltham, MA, USA) for 30 min at 37 °C, 5% CO_2_. Then the slides were washed with PBS 1x and fixed. After rehydration and blocking, the slides were incubated with rabbit polyclonal anti-human VPAC_1_ antibody (1:100, Thermo Scientific, Waltham, MA, USA) and mouse monoclonal anti-human VPAC_2_ (1:50, Abnova, Tapei, Taiwan) 1 h at RT. After washing, Alexa Fluor 488 donkey anti-rabbit IgG and Alexa Fluor 594 goat anti-mouse IgG (1:500, Life Technologies, Waltham, MA, USA) were used as secondary antibodies (1 h at RT). Counterstaining was performed with 1 mg/mL Hoechst. Negative controls were performed in the absence of anti-VPAC_1_ and anti-VPAC_2_ antibodies. Fluorescence was examined on a Leica SP-2 Acousto-Optical Beam Splitter confocal microscope with an inverted stand (Leica DM IRE2; objective, 63X; Leica, St. Gallen, Switzerland). Images were analyzed by ImageJ (Fiji).

### 4.8. Statistical Analysis

Statistical significance was established using a Bonferroni corrected *t*-test analysis, one way-ANOVA, Tukey as a post hoc test and Wilcoxon test (GraphPad Prism 8 Software, San Diego, CA, USA). All the statistical analyses were detailed in each figure.

## 5. Conclusions

Overall, these results show that CD4^+^CD45RA^+^ T cells differentiate, without any Th-lineage polarization microenvironment during the activation of healthy middle-aged donors, mainly to the pathogenic Th17 subset and unpaired Treg subset after several days of activation. Knowing the imprinted profiles of naїve CD4^+^ T cells, which have a pivotal role in the onset and development of autoimmune diseases, it may lead, in healthy and pathogenic conditions, to a new field of study not only in the search for therapeutic targets, but also as a risk or prognostic tool. The presence of VIP in this microenvironment might reduce the Th17 stabilization/amplification and pathogenic Th17/1 profile, confirming the important immunomodulatory role of this neuropeptide also on naїve Th cells. Although further studies are needed to dissect the role of VPAC_1_ and VPAC_2_ receptors in lymphocytes, actual data and previous results show that the alteration of their expression pattern and cellular location linked to changes in lymphocyte status could have important consequences for the lymphocyte response to VIP in non-pathological or different pathological status.

## Figures and Tables

**Figure 1 ijms-23-02346-f001:**
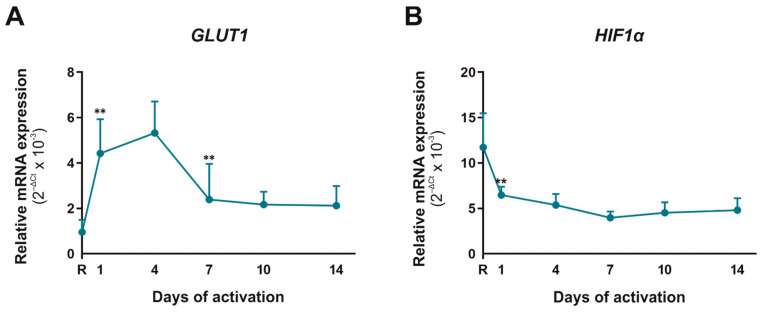
Metabolic biomarkers in human CD4^+^CD45RA^+^ T cells during their activation. mRNA expression of *GLUT1* (**A**) and *HIF1α* (**B**) was determined by semiquantitative real-time PCR analysis in resting condition (R) and after 1, 4, 7, 10 and 14 days of activation with anti-CD3/CD28 beads. Results are expressed as relative mRNA levels (normalized to *SDHA* mRNA levels, 2^−ΔCt^). The mean ± SD of duplicate determination from six different healthy donor samples are shown. Statistical significance was established using one way-ANOVA and Tukey as a post hoc test. Asterisk shows the significance with the previous days of activation (** *p* < 0.01).

**Figure 2 ijms-23-02346-f002:**
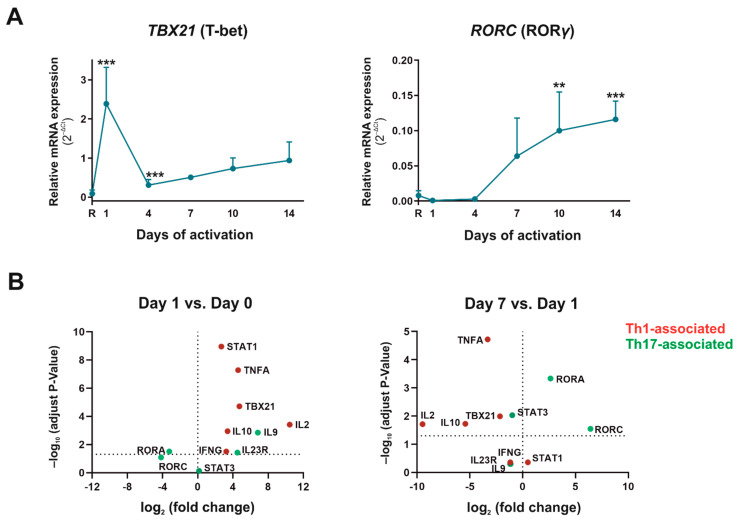
Th1 and Th17 biomarkers in human-activated CD4^+^CD45RA^+^ T cells. (**A**) mRNA expression of *Tbx21* (T-bet)) and *RORC* (RORƴ) was determined by semiquantitative real-time PCR analysis in resting condition (R) and after 1, 4, 7, 10, and 14 days of activation with anti-CD3/CD28 beads. Results are expressed as relative mRNA levels (normalized to *SDHA* mRNA levels, 2^−ΔCt^). The mean ± SD of duplicate determination from six different healthy donor samples are shown. Statistical significance was established using one-way ANOVA in Prism 8 software. Asterisk shows the significance with the previous days of. (**B**) Scatter Dot Plot of mRNA levels of Th profile-associated markers. Left-side: Fold change and *p*-values between resting (day 0) and activated cells for 1 day with anti-CD3/CD28; Right-side: Fold change and *p* values between activated cells for 1 and 7 days. *Y*-axis dot line represents –log_10_ of *p* = 0.05 and *X*-axis dot line represents 0-fold. Up-regulated genes show a positive log_2_ fold change whereas down-regulated show a negative log_2_ fold change. Adjust *p*-values were obtained using a Bonferroni corrected *t*-test analysis in Prism 8 software. Results from data of six different donors measured for replicate are shown. (** *p* < 0.01; *** *p* < 0.001).

**Figure 3 ijms-23-02346-f003:**
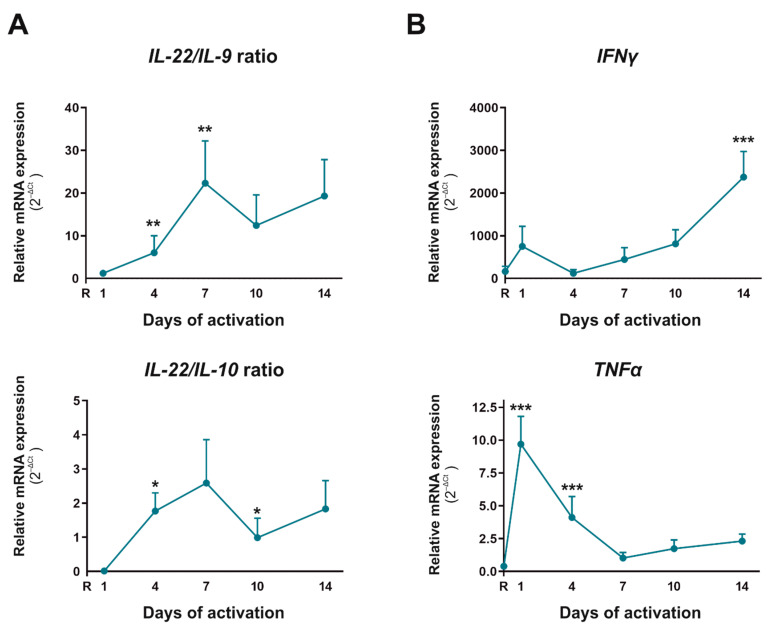
Pathogenic and non-pathogenic Th17 profile in human activated T CD4^+^CD45RA^+^ cells. mRNA expression of cytokines characteristic of a pathogenic-Th17 phenotype (*IL-22*, *IFNγ*, *TNFα*) and non-pathogenic Th17 profile (*IL-9* and *IL-10*) was determined by semiquantitative real-time PCR analysis in resting condition (R) and after 1, 4, 7, 10, and 14 days of activation with anti-CD3/CD28 beads. (**A**) Evolution of the ratio between *IL-22/IL-9* and *IL-22/IL-10*. (**B**) Time-course expression of *IFNγ* and *TNFα*. Results are expressed as relative mRNA levels (normalized to *SDHA* mRNA levels, 2^−ΔCt^). The mean ± SD of duplicate determination from six different healthy donor samples are shown. Statistical significance was established using one way-ANOVA and Tukey as *a* post hoc test. Asterisk shows the significance with the previous days of activation (* *p* < 0.05; ** *p* < 0.01; *** *p* < 0.001).

**Figure 4 ijms-23-02346-f004:**
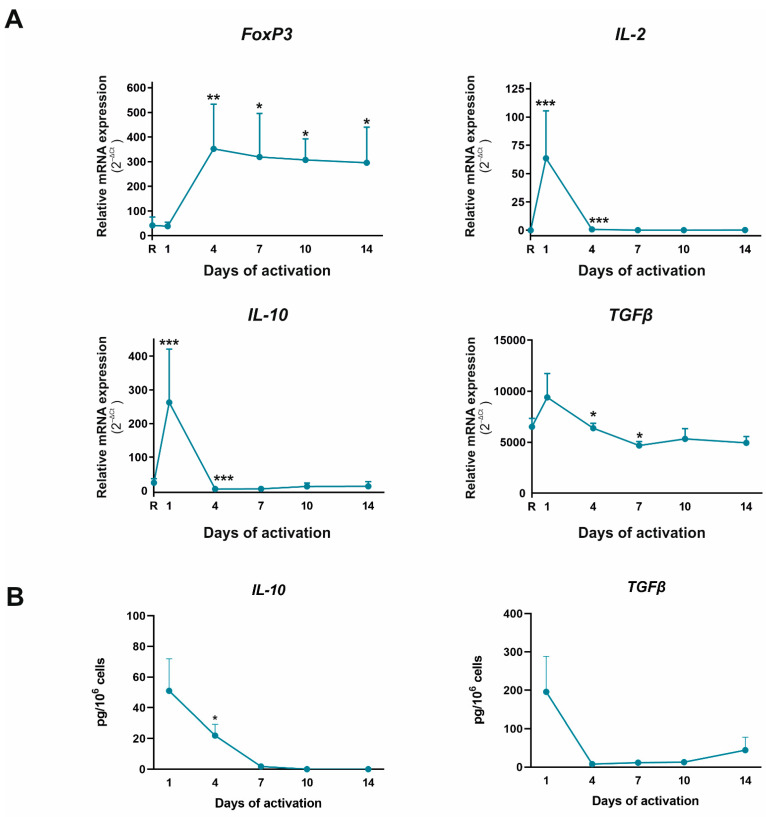
Treg profile in human activated T CD4^+^CD45RA^+^ T cells. (**A**) mRNA expression of *Foxp3*, *IL-10, IL-2,* and *TGFβ* was determined by semiquantitative real-time PCR analysis in resting condition (R), and after 1, 4, 7, 10, and 14 days of activation with anti-CD3/CD28 beads. Results are expressed as relative mRNA expression (normalized to *SDHA* mRNA levels, 2^−ΔCt^). (**B**) Protein expression of IL-10 and TGFβ was measured in culture supernatants by ELISA tests on days 1, 4, 7, 10, and 14. The mean ± SD of duplicate determination from six different healthy donor samples are shown. Statistical significance was established using one way-ANOVA and Tukey as a *post hoc* test. Asterisk shows the significance with previous days of activation (* *p* < 0.05; ** *p* < 0.01; *** *p* < 0.001).

**Figure 5 ijms-23-02346-f005:**
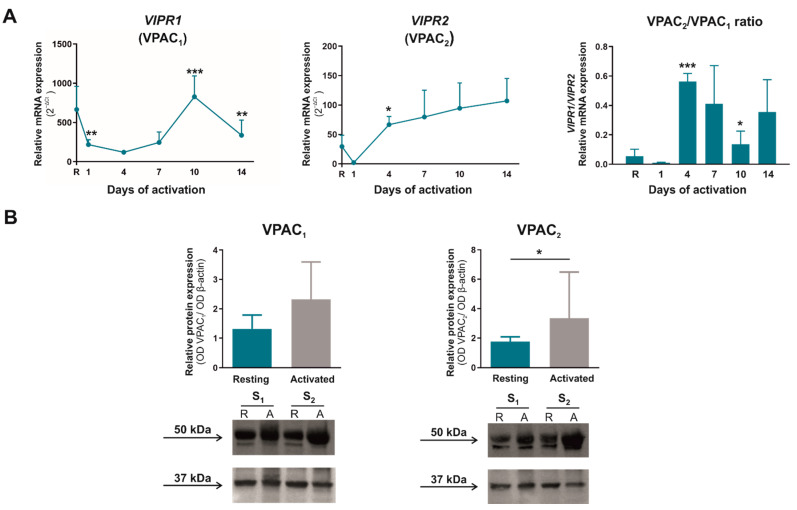
Time course expression of VPAC receptors during the activation of CD4^+^CD45RA^+^
**T cells**. (**A**) mRNA expression of VPAC_1_ and VPAC_2_ was determined by semiquantitative real-time PCR analysis in resting condition (R) and after 1, 4, 7, 10, and 14 days of activation with anti-CD3/CD28 beads. Results are expressed as relative mRNA levels (normalized to *SDHA* mRNA levels, 2^−ΔCt^). The mean ± SD of duplicate determination from six different healthy donor samples are shown. Statistical significance was established using one-way ANOVA and Tukey as a *post hoc* test. Asterisk shows the significance with previous days of activation. (**B**) Protein levels of VPAC_1_ and VPAC_2_ receptors in lysates of resting (R)- and seven days activated (**A**) CD4^+^CD45RA^+^ T cells were measured by Western blotting. β-actin protein levels were determined as a loading control. Protein bands were analyzed by densitometric analysis and normalized against the intensity of β-actin. Statistical significance was established using Wilcoxon paired test. Results represent the mean ± SEM of four different donors. Two representative samples (S_1_ and S_2_) are shown. (* *p* < 0.05; ** *p* < 0.01; ****p* < 0.001).

**Figure 6 ijms-23-02346-f006:**
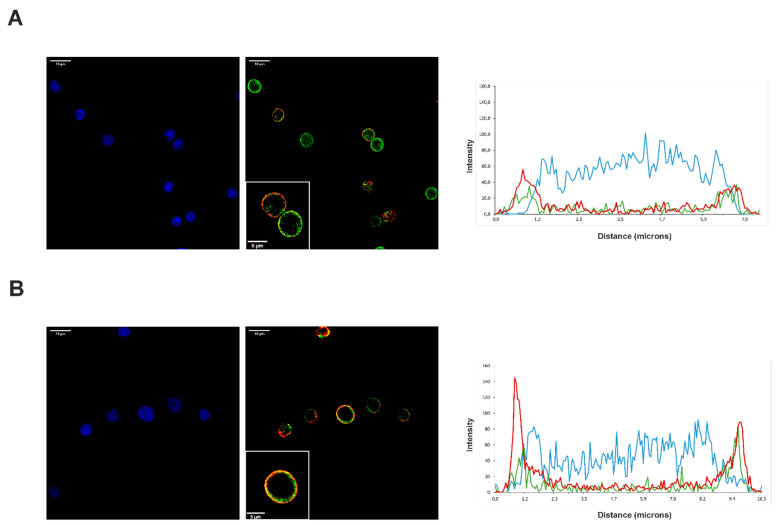
Cellular location of VPAC_1_ and VPAC_2_ in activated CD4^+^CD45RA^+^ T cells. Immunofluorescence analysis showing the distribution of VPAC_1_ (green) and VPAC_2_ (red) receptors with Hoechst counterstaining (blue) across the resting (**A**) and activated (**B**) cells. Boxed areas show higher magnification views of individual cells. Zommed 63 x images were imported into ImageJ to analyze the distribution of VIP receptors across the cells. Representative graphs of intensity profiles plotted as a function distance (measured in microns) versus intensity (measured in RGB-scale values) showed the distribution of receptors on each cell state. A representative experiment of three others is shown. Images were obtained in a confocal microscope, as explained in Materials and Methods. Scale bars represent 10 μm.

**Table 1 ijms-23-02346-t001:** Concentration and Viability of CD4^+^CD45RA^+^ T cells during activation.

	Concentration Cells/mL × 10^6^	Viability %
Day 1	0.19 ± 0.04	64.3 ± 11.2
Day 4	1.2 ± 0.12	81.3 ± 3.8
Day 7	2.53 ± 0.28	74.3 ± 7.8
Day 10	1.75 ± 0.26	50.8 ± 9.8
Day 14	1.15 ± 0.45	28.3 ± 9.4

Measured concentration and viability of CD4^+^CD45RA^+^ T cells after 1, 4, 7, 10 and 14 days of activation with anti-CD3/CD28 beads. The mean ± SD of duplicate determination of four different healthy donor samples are shown.

**Table 2 ijms-23-02346-t002:** Comparative mRNA expression of *IFNγ* with the different treatments.

*IFNγ*	Activated	Activated + VIP	Activated + VPAC_1_ Agonist	Activated + VPAC_2_ Agonist
Resting	166 ± 67.7			
Day 1	750 ± 236	209 ± 58.8 **	420 ± 98.3	489 ± 143
Day 4	122 ± 43.9	33.6 ± 10.5 *	64.4 ± 14.3	60.7 ± 8.0
Day 7	446 ± 124	123 ± 21.9 **	203 ± 15.5 *	175 ± 59.9 *
Day 10	810 ± 148	185 ± 51.9 **	369 ± 102 *	388 ± 102 *
Day 14	2375 ± 244	735 ± 118 **	1097 ± 98.9 *	425 ± 211 *
*IL-23R*	Activated	Activated + VIP	Activated + VPAC_1_ Agonist	Activated+ VPAC_2_ Agonist
Resting	0.011 ± 0.002			
Day 1	0.154 ± 0.079	0.123 ± 0.065	0.074 ± 0.013	0.144 ± 0.076
Day 4	0.111 ± 0.024	0.064 ± 0.009 **	0.075 ± 0.001	0.063 ± 0.016
Day 7	0.136 ± 0.054	0.054 ± 0.008 **	0.100 ± 0.017 *	0.090 ± 0.017 **
Day 10	0.178 ± 0.016	0.115 ± 0.017 **	0.122 ± 0.019	0.119 ± 0.015 **
Day 14	0.157 ± 0.018	0.171 ± 0.014	0.156 ± 0.015	0.205 ± 0.014

mRNA expression of *IFNγ* and *IL-23R* was determined by semiquantitative real-time PCR analysis in resting condition and after 1, 4, 7, 10, and 14 days of activation with anti-CD3/CD28 beads. Results are expressed as relative mRNA levels (normalized to *SDHA* mRNA levels, 2^−ΔCt^). The mean ± SEM of duplicate determination of six different healthy donor samples are shown. Statistical significance was established using Wilcoxon *t* test in Prism 8 software comparing the activated group with the rest of the group (* *p* < 0.05, ** *p* < 0.01).

**Table 3 ijms-23-02346-t003:** Genes analyzed by semiquantitative real-time polymerase chain reaction.

Gene ^1^	GeneBank Accession No. ^2^	Sequence Position/Assay Location (TaqMan^®^) ^3^	Sequence/Assay ID (TaqMan^®^) ^4^
Foxp3	NM_001114377.1	899	Hs01085834_m1
HIF-1α	NM_001243084.1	757	Hs00153153_m1
IFNG (IFNγ)	NM_000619.2	495	Hs00989291_m1
IL-2	NM_000586.3	267	Hs00174114_m1
IL-9	NM_000590.1	325	Hs00914237_m1
IL-10	NM_000572.2	510	Hs00961622_m1
IL-22	NM_020525.4	445	Hs01574154_m1
IL-23R	NM_144701.2	1037	Hs00332759_m1
RORA (RORα)	NM_002943.3	1173	Hs00536545_m1
RORC (RORγt)	NM_005060.3	182	Hs00172860_m1
SDHA	NM_001294332.1	757	Hs00188166_m1
SLC2A1 (GLUT1)	NM_006516.2		Hs00892681_m1
STAT1	NM_007315.3	1832	Hs01013996_m1
STAT3	NM_003150.3	488	Hs01047580_m1
Tbx21 (T-bet)	NM_013351.1	707	Hs00203436_m1
TGFβ	NM_000660.5	1905	Hs00998133_m1
TNFA (TNFα)	NM_000594.3	457	Hs00174128_m1
VIPR1 (VPAC_1_)	NM_004624.3	306	Hs00270351_m1
VIPR1 (VPAC_2_)	NM_003382.4	644	Hs00173643_m1

^1^ Gene, ^2^ genebank accession number, ^3^ sequence position or assay location, and ^4^ sequence or assay ID for each primer used in the study are shown.

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
