# Peer review of "Human CD4+CD45RA+ T Cells Behavior after In Vitro Activation: Modulatory Role of Vasoactive Intestinal Peptide"

_ijms, 2022, doi:10.3390/ijms23042346_

Round 1

Reviewer 1 Report

In the present paper, the authors study the differentiation profile of human activated CD4+CD45RA+ T cells without any preconditioned microenvironment and they try to analyze the role of VIP and its receptors during the activation/differentiation. Although the approach is interesting, and the writing is very careful, the experimental design is poor and makes it very difficult to draw robust conclusions from the study. Therefore, some major revisions are needed before taking the paper into consideration for publication.

-How the purity of CD4+CD45RA+ T cells was confirmed? I guess it was by cytometry. In this case, please provide plots.

-Why did the authors choose SDHA as the reference gene? Using an enzyme involved in the citric acid cycle and mitochondrial respiratory chain to study changes in metabolism doesn't seem very appropriate, does it?

-The authors must show the survival rate of the cells during the 14 days of culture, under the different conditions of activation and also in the presence of VIP and its agonists. Even, additional apoptosis studies would be welcome.

-mRNA expression is not the whole picture about what happens at the protein level. Supernatants must be collected to measure cytokine secretion patterns to further characterize T cell activity. Even, T-cell phenotype should be examined by flow cytometry

Below you can find my minor suggestions

-In the Table 1 foot-“The mean ±”-Please, add SD (line 268)

-Statistical analysis is missing in the materials and methods section. Although this has been described at each Figure legend, it would be convenient to add it to the materials and methods

-Ethical approval number for this study should be included in the Materials & Methods section.

Reviewer 2 Report

Overall, these results show that CD4+CD45RA+ T cells differentiate, without any Th- lineage polarization microenvironment during the activation of healthy middle-aged do- nors, mainly to pathogenic Th17 subset and to unpaired Treg subset after several days of activation.

Knowing the imprinted profiles of naïve CD4+T cells, which have a pivotal role in the onset and development of autoimmune diseases, it may lead, in healthy and pathogenic conditions, to a new field of study not only in the search for therapeutic targets, but also as a risk or prognostic tool

The presence of VIP in this microenvironment might reduce the Th17 stabilization/amplification and the pathogenic Th17/1 profile, confirming the important immunomodulatory role of this neuropeptide, also on naïve Th cells.

Although further studies are needed to dissect the role of VPAC1 and VPAC2 receptors on lymphocytes, actual data and previous results show that the alteration of their expression pattern and cellular location linked to changes in lymphocyte status could have important consequences for the lymphocyte response to VIP in non-pathological or different pathological status.

Author Response

We sincerely appreciate the comments of reviewer 2. There are no questions or issues for improvement our manuscript under his review.

Round 2

Reviewer 1 Report

I thank the authors for all the clarifications and the effort to follow my recommendations in order to improve the article.

However, from my point of view, there are still some details that could be improved.

As far as I can understand from the materials and methods, the authors measured cell viability, not proliferation. Viability and proliferation are two distinct characteristics of cells. Viability is a measure of the number of living cells in a population whereas proliferation is a measure of cell division. It should be noted that not all viable cells divide. Please correct the manuscript using the appropriate term so that there is no confusion on the part of the reader.

On the other hand, the authors note a loss of viability from day 7 onwards, so the changes seen in cytokines from this day onwards may be due to this reason and it is important to be discussed in the text. I invite the authors to give a little more thought to their results, especially considering the great work they already did in the discussion.

This reviewer recommended a phenotypic analysis of the subpopulations that would confirm, without a doubt, to which lineage the cells naturally polarize. The authors made no reference to this comment.

Finally, the authors did not provide the ethics committee approval reference number.

Round 3

Reviewer 1 Report

This reviewer thanks the authors for their promptness and willingness to answer and attend to the suggestions made. The article shows dedication and this reviewer appreciates it. So I am sure the authors will want the final version to be perfect. By virtue of this, this reviewer still has a few comments in relation to author’s responses.

  1. I believe there is a mistake in the table 1 regarding % of viability. For example: if at day 1, 190.000 live cells/mL represent 64%, that means that there were about 300.000 cells at day 0. Then, at day 4, 1.2 x 106 cells cannot be 81%. Please review all the data and correct where would be necessary.
  2. If the cytokine data in culture supernatant are normalized to the number of live cells, it is important to clarify this in the text or in the figure legend, since as it is now put it is interpreted as the number of cells in the initial culture. This can be done, as long as viability is preserved, which is not the case. Another way to represent it is as a function of protein concentration. Regarding the mRNA expression data, it is true that they are normalized to the reference gene expression, which is correct. What this reviewer meant is that the loss of viability of certain cell types can condition the behaviour of the others and alter the pattern of cytokine expression. Hence the importance of verifying whether there are changes in cell viability and proliferation.
  3. Analyzing the expression pattern of cytokines is not sufficient to characterize the subpopulations of the immune system since cytokines are produced by many cell types. For example, IL-22 is not exclusive to Th-17 cells, as Th1 and Th22 cells can also produce IL-22.
  4. All projects that request approval from the ethics committee obtain an approval reference number. The project under which this article is framed must have it and it must be reflected in the manuscript.